# Chirality-Dependent and Intrinsic Auxeticity for Single-Walled Carbon Nanotubes

**DOI:** 10.3390/ma15248720

**Published:** 2022-12-07

**Authors:** Hai-Ning Zhang, Yin Fan, Hui-Shen Shen

**Affiliations:** School of Aeronautics and Astronautics, Shanghai Jiao Tong University, Shanghai 200240, China

**Keywords:** carbon nanotubes, chirality, Poisson’s ratio, molecular dynamics, auxetic, rigidity

## Abstract

Single-walled carbon nanotubes (SWCNTs) have superior mechanical properties which originate from a strong C-C covalent bond and unique nanostructure. Chirality, one of the helical structural parameters of SWCNTs, leads to differences in mechanical performance. In this work, molecular dynamics (MD) simulation was performed to analyze engineering Poisson’s ratio (EPR) and incremental Poisson’s ratio (IPR) of SWCNTs with different chiral angles, respectively, under tensile and compressive load, as well as the chiral effect on rigidity. We reported the minimum EPR for (4, 1) SWCNT and obtained the distribution and trend of EPR which is dependent on chiral index *m*. In addition, a new observation showed two exactly opposite trends of EPR existing not only in tension and compression but also in the longitudinal and radial directions. Furthermore, we found that the critical strain, over which SWCNT would be auxetic, ranged from 6% to 18% and was also chirality-dependent. Three representative SWCNTs with chiral angle of 0° (zigzag), 10.89° (chiral), and 30° (armchair) were selected for the mechanism study of auxeticity. Finally, a method of the contribution to radial strain for two main deformation modes proposed in this paper could well explain the negative IPR phenomenon.

## 1. Introduction

The discovery [1] of carbon nanotubes (CNTs) has accelerated exploration of the nano area and nowadays nanotechnologies are changing our world. Due to excellent mechanical properties, CNTs, especially for single-walled carbon nanotubes (SWCNTs), are regarded as an ideal candidate for reinforcement of next-generation advanced composite materials [2]. Hence, a better understanding of the mechanical performance of CNTs is key issue of future applications in engineering and industrial materials. Among all mechanical parameters, Poisson’s ratio is an important mechanical indicator, reflecting the response of a material subjected to external loads. Conventionally, the value equal to the negative ratio of radial strain to axial strain is normally defined as engineering Poisson’s ratio (EPR, *υ*) of CNT, while incremental Poisson’s ratio (IPR, *υ ^in^*) describing variation of material EPR at some time is the value of the negative ratio of radial differential strain to axial differential strain. In addition, a reasonable and accurate characterization of rigidity (*C*), which reflects the resist deformation ability, is necessary for application of CNTs as a candidate for reinforcement.

Molecular mechanics (MM) paved an effective way to study EPR of SWCNTs. In the MM model, the C-C bond in CNTs is always treated as a beam of nonlinear spring structure [3]. However, the predicted results of SWCNT EPR from different research teams seem inconsistent, even though they were all based on MM, and without considering chirality. For example, Zhu and Wang [4] estimated that the EPRs of the zigzag and armchair SWCNTs with a big enough diameter would both reach about 0.27, which is approaching to the reported results [3,5,6,7,8]. Meanwhile, the EPRs of the SWCNTs (with diameter > 2 nm) calculated by Xiao et al. [9] were close to 0.21, which is very close to a lattice-dynamics result [10]. Other researches [11,12,13] indicated that the EPR of large-diameter SWCNT would go down to 0.16, that is also the EPR of graphite sheet [14]. The main reason for these differences is because of various interaction between carbon atoms. As is well known, chiral SWCNTs are those excepting zigzag and armchair SWCNTs, and chirality is represented by indices (*n*, *m*) which determines both size and chiral angle of a SWCNT. Some of above studies also reported that ERPs of chiral SWCNTs were between those of armchair SWCNTs and zigzag SWCNTs. On the other hand, EPRs of CNTs can be expressed in analytical formulas which are independent on deformation through MM modelling, therefore, IPRs of SWCNTs seem impossible to be studied.

On this problem, molecular dynamics (MD) can effectively predict historical EPRs which is adaptive to a further analysis of IPRs [15,16]. Based on a second-generation reactive empirical bond order (REBO) potential [17], Ni et al. [18] obtained variations of EPRs of both zigzag and armchair SWCNTs during both tensile and compressive processes. The compressive and tensile simulation of chiral SWCNTs was performed by Shintani and Narita [19], and they found chiral-dependence could be in proportion to the magnitude of strain. Especially, both Ni et al. [18] and Shintani and Narita [19] reported that EPRs show thoroughly reversed trends, respectively, under tensile and compressive load. Recently, Canadija [20] proved that a deep learning neural network developed on the basis of MD results could accurately predict mechanical properties (including EPR) of SWCNTs. Although IPR can be calculated from MD simulation, there have not been any SWCNT IPR results found from the existing literature, as far as the authors can find. Whereas another allotrope of carbon, graphene, was reported to have negative IPR at the tensile strain of about 6% [21]. So, we guess the IPR of SWCNTs would also turn out to be negative in the tension process and the turning point is chirality-dependent.

In mechanics, there is a controversy on how to model SWCNT following continuum theory. For instance, Bian et al. [3] treated CNTs as an isotropic material in their MM model, as they thought effective wall thickness (*h*) of CNTs was too small and thus neglected the radial stress. This assumption was similar to that in the study of Zhu and Wang [4]. Nevertheless, some other researchers [11,22] have noted that the effective elastic properties of a SWCNTs were permanently transversely isotropic. Compared with work of Chang et al. [22] where only two force constants were used, Li and Guo [23] adopted four force constants to describe C-C bonds and claimed that SWCNTs were prone to degenerate from transversely isotropic to isotropic if the diameter of SWCNTs turned out to be larger. A similar conclusion was also reported by Refs. [13,24]. More specifically, Li and Chou [25] conducted MM method on SWNTs under hydrostatic pressure and found that radial Young’s modulus (*E_r_*) was not equal to circumferential Young’s modulus (*E_θ_*). Furthermore, studies employing the finite element (FE) model also gave rise to variable conclusions. Pereira et al. [26] considered that it was advisable to calculate Poisson’s ratio assuming isotropy. Jalan et al. [27], through FE analysis, obtained the five transversely isotropic properties of SWCNTs, while Ghavamian et al. [28] thought that CNTs behave rather as an anisotropic material.

On the other hand, determination of the effective thickness of the SWCNT “wall” in mechanics is still a problem. To calculate Young’s modulus of SWCNTs, one usually needs to define the effective wall thickness based on continuum elasticity theory. However, the structure of SWCNTs is inherently discrete [29] and the “wall” of a SWCNT consisting of several carbon atoms practically do not form the continuous spatial distribution [30]. Hence, the assumption of the effective wall thickness whose values range from 0.617 Å [31] to 6.9 Å [32] in part results in the dispersion of Young’s moduli [30,33,34,35]. Many researchers [4,6,7,8,9,20,33,36,37] took the interplanar spacing of graphite layers 3.4 Å as the effective wall thickness, while some other researchers [38,39] considered that the effective wall thickness is equal to the extent of π orbitals in the out-of-plane direction. Through the calculation of the flexural rigidity *Eh*^3^/12(1 − *υ*^2^) and axial rigidity *C* = *Eh*, the thickness of 0.66 Å and Young’s modulus about 5.5 TPa were reported by Yakobson et al. [38].In addition to choosing the value that possesses specific physical meaning, researchers [31,32,40,41] also tried many different ways to obtain the effective wall thickness. For instance, Vodenitcharova and Zhang [31] used the ring theory continuum mechanics model to predict the Young’s modulus about 4.88 TPa and thickness of 0.617 Å, which was cited by Mylvaganam and Zhang [42]. Huang et al. [41] linked the thickness directly to the interatomic potential and found that the thickness depends on the choice of potential, type of loading, the diameter and chirality of CNTs. Odegard et al. [32] obtained the thickness of 6.9 Å by equating the molecular potential energy with the strain energy stored in the equivalent truss and continuum model.

In fact, Fan and Shen [43] have already claimed that hypothesis of continuum mechanics cannot exactly predict mechanical characteristics of nanomaterials, although some concepts from continuum mechanics we have employed to analyze their mechanical behaviors. Thus, to avoid the confused assumption of *h*, we use the ratio of rigidity as a substitute for analysis of the ratio of Young’s moduli. Obviously, ratio of rigidity in the longitudinal direction (*C_l_*) to rigidity in the radial direction (*C_r_*) equals to the ratio of the longitudinal Young’s modulus (*E_l_*) to radial Young’s modulus (*E_r_*), namely *C_l_*/*C_r_* = *E_l_ h*/*E_r_ h* = *E_l_*/*E_r_*. To the best of the authors’ knowledge, this will be the first time to address this problem specially.

To fully understand the chiral effect on Poisson’s ratios and ratios of moduli of SWCNT, we implemented an MD simulation to analyze EPRs, IPRs, and ratio of moduli of SWCNTs subjected to tensile or compressive load. Particularly, we engaged in two types of comparison: longitudinal vs. radial properties; tensile vs. compressive results. The adaptive intermolecular reactive empirical bond order (AIREBO) [44] potential is used for the force field between carbon atoms. The auxetic phenomena as negative IPRs in chiral SWCNTs will be discussed in detail. The present work not only reveals chiral-dependent insights of mechanical properties of SWCNTs, but also first discovered the auxetic phenomenon of perfect SWCNT. It is believed that with these new findings, our work will contribute to future study and application of SWCNTs [45,46].

## 2. Models and Methodology

As shown in Figure 1a, a cylindrical SWCNT can be imagined as a rolled graphene sheet where the two crystallographically equivalent points, O and A, coincide. The chiral vector ***C_h_*** that starts from point O and ends on point A, can be expressed in terms of the basic vectors ***a***_1_ and ***a***_2_ as:(1)$$C_{h}=na_{1}+ma_{2}\equiv (n,\;m),$$
where (*n*, *m*) are chiral indices that define helical structural parameters of a SWCNT, i.e., diameter *d* and chiral angle *θ*. Chiral angle can be theoretically calculated by
(2)$$\theta =\cos^{-1}\left(\frac{2n+m}{2\sqrt{n^{2}+\mathrm{nm}+m^{2}}}\right),$$

It should be noted that *θ* is the angle between the chiral vector ***C_h_*** and the unit vector ***a***_1_, which varies from 0° to 30° when *n* ≥ *m*. Specially, chiral angle 0° refers to “zigzag” SWCNT (*m* equals to zero), while a chiral angle of 30° refers to “armchair” SWCNT (*m and n* are equal). The rest of SWCNT are regarded as “chiral”, as shown in Figure 1b.

Due to its tubular structure, transverse deformation of a SWCNT can be calculated by variation of tube radius. When a SWCNT is stretched along the longitudinal (or radial) direction, the EPR and IPR of the SWCNT are, respectively, calculated as:(3)$$\upsilon_{12}=-\frac{\varepsilon_{r}}{\varepsilon_{l}};\;\upsilon_{21}=-\frac{\varepsilon_{l}}{\varepsilon_{r}},$$
(4)$$\upsilon_{12}^{\mathrm{in}}=-\frac{d\varepsilon_{r}}{d\varepsilon_{l}};\;\upsilon_{21}^{\mathrm{in}}=-\frac{d\varepsilon_{r}}{d\varepsilon_{r}},$$
where *ε_l_* and *ε_r_* are strains along the longitudinal and radial directions, respectively. Likewise, subscripts 1 and 2 refer to longitudinal and radial directions, respectively. In the MD simulation, however, the circles constructed by carbon atoms in any arbitrary section of a SWCNT can never be perfect in the simulation because of molecular vibration. If we do not consider the vibration effect, any atom in a section can be found its symmetric atom about circle center. Assuming that there are *N* atom pairs in a section, the equivalent radius $\bar{r}$ can be seen as the half of the mean value of the distances for these atom pair and expressed as:(5)$$\bar{r}=\frac{1}{2N}\sum_{1}^{N}d_{\mathrm{ij}},$$
in which *d_ij_* denotes the distance between the atom *i* and the atom *j*, that are in a pair.

The rigidity of SWCNTs subjected to tension or compression is derived from force approach as:(6)$$C_{11}^{T}=E_{11}^{T}h=-\frac{\sigma_{{}_{l}}^{T}}{\varepsilon_{{}_{l}}^{T}}h,\;C_{22}^{C}=E_{22}^{C}h=-\frac{\sigma_{{}_{r}}^{C}}{\varepsilon_{{}_{r}}^{C}}h,$$
where superscripts T and C stand for tension and compression, respectively. *σ_l_* is stress along the longitudinal direction evaluated by *σ_l_* = −*S_l_*/*V*_0_. Here, *V*_0_ is the volume of a SWCNT determined as *V*_0_ = 2*π*$\bar{r}$*hL* where *L* is the length of a SWCNT. *S_l_* is the stress tensor [47] composed of kinetic energy component and virial component and defined as follows:(7)$$S_{\mathrm{ab}}=-mv_{a}v_{b}-W_{\mathrm{ab}},$$
(8)$$W_{\mathrm{ab}}=\frac{1}{2}\overset{N_{p}}{\underset{k=1}{\sum }}(r_{I_{a}}\cdot F_{I_{b}}),$$
in which subscripts *a* and *b* represent the direction of *x*, *y*, *z* in Cartesian coordinates, and *I* refers to individual atom. *v* is the velocity component of atom *I* and hence *v_a_*·*v_b_* represents kinetic energy contribution. The second term *W_ab_* is the virial contribution due to intra and intermolecular interactions, where *k* loops over the *N_p_* neighbors of specified atom *I*. Term ***r*·*F*** is the dot product between position vector ***r*** and force vector ***F*** resulting from the pairwise interaction on specified atom, i.e., projection of ***F*** on the position vector ***r*** direction. Therefore, radial stress tensor (if decomposed in *x*-*y* plane perpendicular to the CNT axis) can be expressed as:(9)$$S_{r}\approx -m(v_{x}^{2}+v_{y}^{2})-\frac{1}{2}\overset{N_{p}}{\underset{k=1}{\sum }}r_{I}\cdot (F_{I_{x}}+F_{I_{y}}),$$

The choice of the potential is known to have crucial consequences for the results of calculated properties because each potential has its drawback. In the present case, if there is no other statement, the AIREBO [44] potential with a cutoff radius (3 Å) is selected to describe carbon–carbon interactions, which is widely used to study SWCNTs [20,37,48,49,50,51]. The definition of AIREBO potential consists of three terms [52,53]:(10)$$E^{\mathrm{AIREBO}}=\frac{1}{2}\sum_{i}\sum_{j\neq i}\left[E_{\mathrm{ij}}^{\mathrm{REBO}}+E_{\mathrm{ij}}^{\mathrm{LJ}}+\sum_{k\neq i,j}\sum_{g\neq i,j,k}E_{\mathrm{kijg}}^{\mathrm{TORSION}}\right],$$
where subscripts *i*, *j*, *k*, and *g* refer to individual atom. It is clear that the AIREBO potential considers non-boned interactions $E^{LJ}$ and torsional interactions $E^{TORSION}$ besides REBO terms, which is a short-range model describing a C–C atomic interaction with a fixed cut-off range of 2.0 Å. In Equation (10), $E_{ij}^{REBO}$ is given as:(11)$$E_{\mathrm{ij}}^{\mathrm{REBO}}=f(r_{\mathrm{ij}})(V_{\mathrm{ij}}^{R}+B_{\mathrm{ij}}V_{\mathrm{ij}}^{A}),$$
where $B_{ij}$ is a bond-dependent parameter which weighs the bond order, $V_{ij}^{R}$ and $V_{ij}^{A}$ are repulsive and attractive pair energy terms, respectively, *f* (*r_ij_*) is the cut-off function of the distance *r_ij_* between atoms *i* and *j* [17,54].

The simulations were carried out using open-source molecular dynamics software LAMMPS [55] (large-scale atomic/molecular massively parallel simulator). Periodic boundary conditions are only set in the longitudinal direction, so that the length of the simulated SWCNT can be seen as infinitely long. In the meantime, we make it possible to remove the effect of discrepancy of length with respect to a varied chiral angle [56]. The simulation box boundaries, apart from longitudinal direction, are free to shrink. The duration of each timestep is 0.5 fs [42] and the temperature of system is set and maintained 4.3 K by employing the Nosé–Hoover thermostat to minimize the impact of randomization of C atoms thermal fluctuations. The minimized structure is then equilibrated for a period of 10^5^ timesteps. Following equilibration, stretching of SWCNTs follows a specified constant engineering strain rate of 2 × 10^−5^/ps with 10^6^ timesteps in the canonical ensemble (NVT). Additionally, to obtain the transverse properties, the movement of each atom in a radial direction is controlled by the command *addforce* with 10^6^ timesteps, which applied a time-varying force increment based on each atom coordinate [57]. When performing the compression test, engineering strain rate and *addforce* are set to be negative.

## 3. Results and Discussion

### 3.1. Engineering Poisson’s Ratio (EPR) under Tension

The longitudinal strain of SWCNTs for each timestep can be directly obtained from the simulation, while the corresponding transverse strain can be calculated by means of equivalent radius. Then, an estimation from the linear variation of the transverse strain within 0.5% longitudinal strain is performed for EPR *υ*_12_. In contrast, EPR *υ*_21_ is calculated through a linear fitting procedure within 0.5% transverse strain when SWCNTs expand along the radial direction. We simulated a series of 240 SWCNTs with diameter *d* ranging from 3.59 Å for the (4, 1) SWCNT [58] to 119.85 Å for the (150, 6) SWCNT and their EPRs $\upsilon_{12}^{T}$ and $\upsilon_{21}^{T}$ with different chiral indices *m* (between one and six) are shown in Figure 2a and Figure 2b, respectively. It can be observed that both $\upsilon_{12}^{T}$ and $\upsilon_{21}^{T}$ are varied regularly with chiral angle when *m* is fixed, in other words, *m* decides the trend of EPR. As regards $\upsilon_{12}^{T}$, however, this variation is not so remarkable when *m* is over three. From another perspective, the increase of index *n* also declines the insensibility of EPR to *m*. Furthermore, another important finding from Figure 2a is that the EPR curves with the same *m* would converge to one point at zero chiral angle, that is to say, EPR of zigzag SWCNT is chiral-independent and its value is fixed at about 0.29, which is very close to a reported MD result [59]. Figure 2b reveals that $\upsilon_{21}^{T}$ also converge to a value near 0.33, surpassing $\upsilon_{12}^{T}$. Comparatively, EPRs $\upsilon_{21}^{T}$ of each group of same chiral index *m* reduce more pronouncedly and keep sensible to *m*.

EPRs variation with respect to the diameter is illustrated in Figure 3 with the aim of comparing the influence of chiral angle and diameter. Figure 3 uncovers that EPRs are positively associated with the diameter and more approaching SWCNTs with larger diameters [18]. As in conjunction with Figure 2, it can be inferred that SWCNTs are influenced by the combined function of chiral angle and diameter when the diameter is less than 20 Å. Chiral index *m* seems to affect more $\upsilon_{12}^{T}$ than $\upsilon_{21}^{T}$. For diameters above 20 Å, EPRs are almost constant and chiral angle becomes the prevailing factor, in close agreement with what was stated in Ref. [20]. Based on the above observations, we selected four groups with diameters of 10 Å, 15 Å, 20 Å and 25 Å to study the effect of the chiral angle separately in the following section. For convenience, we decide to represent them as G25, G20, G15 and G10, respectively.

To explain these phenomena, the corresponding initial potential energy (IPE) of 240 SWCNTs is depicted in Figure 4. It is evident that the distribution and trend of IPE is similar to those of EPR in Figure 2 and Figure 3. The IPE reflects the SWCNT capacity on resistance of axial deformation. In other words, the higher IPE means the stronger the resistance capacity to deformation; as a result, EPR is smaller. Although IPE accounts for the overall changes of EPR with chiral angle and diameter satisfactorily, IPE is not the cause of the inconsistency between *υ*_12_ and *υ*_21_, given that each configuration (*n*, *m*) possesses only one IPE. Actually, IPE just reflects the stationary state of SWCNTs and cannot describes dynamic processes. The discrepancy has essentially arisen from the performance variations among zigzag and armchair structure, which has been reported not only in SWCNTs [18,19,20,37] but also in graphene [16,60]. It also shows that chirality-dependence is not negligible [14,49,61].

Theoretically, we can construct various SWCNTs with minimum diameter according to different chiral indices *m*. The EPRs of the SWCNTs, whose diameter is the smallest for a certain *m*, are listed in Table 1. It is worth noting that (4, 1) SWCNT possesses the smallest diameter in theory and its EPR reaches the value of 0.0385 for $\upsilon_{12}^{T}$ and 0.0284 for $\upsilon_{21}^{T}$ which is also the smallest value we can obtain in theory.

Next, the $\upsilon_{12}^{T}$ of the SWCNTs with fixed 10.89° chiral angle but different diameters are compared in Table 2. The comparison between different potentials of AIREBO and REBO is also listed in Table 2. Owing to lack of items of $E^{LJ}$ and $E^{TORSION}$ [62], the $\upsilon_{12}^{T}$ for REBO is higher than those for AIREBO. The absolute value of $\upsilon_{12}^{T}$ increases with the enlargement of diameter uniformly for 2 potentials. However, on the rising trend of EPR for a larger diameter, they are the same.

### 3.2. EPR and Ratio of Young’s Moduli under Tension and Compression

As aforementioned, the increase of chiral index *m* or *n* would reduce the variation range of EPR, which is concluded from Figure 2. Since increasing *m* or *n* leads to the same result of larger diameter, what happens if the diameter of SWCNT is enough big? A conclusion form Figure 3 is that the *υ*_12_ and *υ*_21_ would infinitely approach the values of 0.29 and 0.33, respectively, when the SWCNT is large enough. In fact, the IPE has a similar trend shown in Figure 4b, which can explain the conclusion. Moreover, the sensibility of EPR to index m is further decreased and the dependence is mainly on the chiral angle. Moreover, very few researches have made the comparison between *υ*_12_ and *υ*_21_ with load type (tension and compression). Hence, we collect and plot EPRs of SWCNTs with diameter of 20 ± 1 Å subjected to tension or compression separately in Figure 5a, which encompasses 40 configurations. It can be seen that the sensibility of EPR to index *m* is further decreased and the dependence is mainly on chiral angle. There are two obvious trends exhibited in Figure 5a: (1) For tensile deformation, as the chiral angle increases from zero degree (zigzag) to thirty degrees (armchair), $\upsilon_{12}^{T}$ increases, while $\upsilon_{21}^{T}$ decreases. Armchair SWCNTs have bigger $\upsilon_{12}^{T}$ but smaller $\upsilon_{21}^{T}$ than zigzag ones. The former academic view is also reported by studies [18,20,35,61]; (2) In contrast to tension, SWCNTs with compressive deformation display the exact opposite behavior [18,19]. The $\upsilon_{12}^{C}$ decreases, while $\upsilon_{21}^{C}$ increases when the chiral angle is increased. Thus, the highest $\upsilon_{12}^{C}$ and lowest $\upsilon_{21}^{C}$ are observed in zigzag SWCNTs, which is in close agreement with Ref. [59]. In addition, EPRs subjected to compression is generally larger than tension.

Note that the ratio of Young’s moduli *E*_11_ to *E*_22_, equal to ratio of rigidities *C*_11_ to *C*_22_, is derived from the Equation (6), namely *E*_11_/*E*_22_ = *C*_11_/*C*_22_. The rigidity *C*_11_ and *C*_22_ are calculated by Equations (7) and (9), using the output parameter stress tensor *S* of LAMMPS, which is really a <stress*volume> formulation. Figure 5b depicts the change of ratio of Young’s moduli *E*_11_/*E*_22_ with chiral angle. It can be clearly observed that *E*_11_/*E*_22_ shows the similar dependence on the chiral angle to EPRs and the opposite trend under different loading type (tension and compression). Ratio of Young’s moduli reduces from larger than 1 to less than 1 as the chiral angle increases under tension, which means that only a few configurations match *E*_11_/*E*_22_ = 1, i.e., *E*_11_ = *E*_22_ (the deviation within −0.4%~0.4%). Particularly, these configurations labeled in green in Figure 5a,b whose chiral angle is 18.38° and 19.11° also satisfy *υ*_12_ = *υ*_21_. EPRs and rigidity of these distinct SWCNTs under different loading type are presented in Table 3 and Table 4 with other three groups G25, G15, G10 chosen in Figure 3. There are also two chiral angles marked in red representing the intersection of two fitting lines (*υ*_12_ and *υ*_21_) or the intersection of fitting line and *x*-axis (Y = 1). Apparently, the above chiral angles are only in theory and do not correspond to integral chiral indices (*n*, *m*). In addition, the loading type has little influence on the intersection. It should be pointed out that when chiral angle is before the intersection, $\upsilon_{12}^{T}$ is less than $\upsilon_{21}^{T}$ and $E_{11}^{T}$/$E_{22}^{T}$ is greater than 1, which means most SWCNTs do not meet Maxwell’s theorem, i.e., *E*_11_
*υ*_21_ = *E*_22_
*υ*_12_. Results of SWCNTs subjected to compression also indicate the above observation. Thus, it is inappropriate to treat all SWCNTs as transversely isotropic or simply orthogonal anisotropic material.

### 3.3. Incremental Poisson’s Ratio (IPR) under Tension

IPR, which reflects the variation of EPR in stretch, can be obtained from the historical EPR curve. We investigate the three IPRs $\upsilon_{12}^{in}$ of the chiral SWCNTs with chiral angle of 0° (zigzag), 10.89°, and 30° (armchair) in Figure 6a, and find that they would drop to be negative at the strain of 5.93%, 10.60%, and 17.33%, respectively. We notice that Jiang et al. [21] predicted the IPR of the zigzag graphene would turn to be negative at about 6% tensile strain, which is in perfect agreement with our simulation result of the (25, 0) zigzag SWCNT. Our new finding is that the (24, 6) chiral and (14, 14) armchair SWCNTs have a different critical strain, which is the turning point of IPR from positive to negative. To further study the effect of chirality on critical strain, a group of 68 SWCNTs with various chiral angles from zero degree to thirty degrees are selected to be stretched by MD simulation and their critical strain $\varepsilon_{11}^{cr}$ and $\varepsilon_{22}^{cr}$ is shown in Figure 6b. If we take 15° chiral angle as a dividing line (green dashed line), the rise of $\varepsilon_{11}^{cr}$ is remarkable at the left of the line, but the trend is extremely slowed at the right. Conversely, $\varepsilon_{22}^{cr}$ increases slightly as the chiral angle increases until 13.37°, corresponding to configuration (22, 7), and then drops sharply to 6% at chiral angle 30° (armchair). Both minimum values of $\varepsilon_{11}^{cr}$ and $\varepsilon_{22}^{cr}$ are nearly 6%. The intersection of two critical strain fitting lines is about 15.9°. Furthermore, an interesting finding is that parts of $\varepsilon_{11}^{cr}$ (0~10°) and $\varepsilon_{22}^{cr}$ (20~30°) are approximately symmetric about the 15° axis.

Numerical previous studies [5,9,11,63,64,65,66,67] proved that bond stretching and angle variation are two main deformation modes affecting Poisson’s ratio. By assessing the two deformation modes, Jiang et al. [21] well explained the auxetic phenomenon of the graphene and the corresponding critical strain. The situation, however, is more complex in SWCNT because of its geometric characteristic with curvature. It should be mentioned that bonds and angles are both varying in the stretch and their effect on IPR, therefore, is intercoupled. So, the auxetic mechanism can only be found by comparing their independent contributions to IPR. According to the definition of IPR in Equation (4), radial strain differential directly determines its plus or minus. Then, the problem is simplified to find the radial strains respectively induced by bond stretching and angle variation. To fully analyze the influence of the variation of bond length and angle on radial strain, a representative unit consisting of four carbon atoms for SWCNT is proposed in Figure 7a. In that unit, there are three bonds with length of *a*, *b*, and *c* and three angles formed by these bonds. It is worth mentioning that *a*, *b*, and *c* are different when the SWCNT is stretched. In a chiral SWCNT, an arbitrary selected section should include *n* bonds of length *a*, *m* bonds of length *b*, and (*n* + *m*) bonds of length *c*. Since the projections of these bonds in *x*-*z* plane can construct a roughly circle, we can derive the following equation:(12)$$2n\left(arcsin\frac{\bar{a}}{2r}\right)+2m\left(arcsin\frac{\bar{b}}{2r}\right)+2(n+m)\left(arcsin\frac{\bar{c}}{2r}\right)=2\pi ,$$
in which $\bar{a}$, $\bar{b}$, and $\bar{c}$ are the projective lengths corresponding to the bonds with length *a*, *b*, and *c*, respectively, and expressed as:(13)$$\bar{a}=a\;\cos(\frac{\alpha }{2}-\frac{\pi }{6}+\theta ),$$
(14)$$\bar{b}=b\;\cos(\frac{\alpha }{2}+\frac{\pi }{6}-\theta ),$$
(15)$$\bar{c}=c\;\cos(\frac{\pi }{6}-\theta ).$$

The contributions to radial strains can be obtained by solving Equations (12)–(15) using Newton’s method.

Figure 7 illustrates the radial strain for three SWCNTs, respectively, induced by bond stretching and angle variation, as well as their IPRs. It is found from Figure 7 that bond stretching leads to positive radial strain resulting in negative Poisson’s ratio, while angle variation contributes negative radial strain resulting in positive Poisson’s ratio. Likewise, Jiang et al. [21], through very simplified algebra, obtained Poisson’s ratio for bond stretching as −1/3 and for angle variation as 1. With the increase of chiral angle, the contributions to radial strain from bond stretching and angle variation are both increased. Their corresponding contributions to IPR, shown in Figure 7b, reveal the reason of critical strain variation for different SWCNTs. For example, the IPR contributed by angle variation for the (25, 0) SWCNT is dominant at the beginning, but its declining rate is higher than that contributed by bond stretching. The intersection point for the two curves is at the tensile strain of 5.85%, which is a good match with the critical strain (5.93%) for the same SWCNT depicted in Figure 4a. After that, the effect of bond stretching is dominant instead, the auxetic phenomenon, therefore, appears, i.e., IPR turns out to be negative. Following this way, the cases of (24, 6) and (14, 14) SWCNTs can also be successfully explained. In summary, the critical strain for negative IPR depends on which mode (bond stretching or angle variation) is dominant.

Tensile behavior curves of (24, 6) chiral SWCNT IPR under different temperatures (4.3 K, 300 K, and 500 K) are plotted in Figure 8a. The remarkable drop of IPR is observed with temperature increase when the tensile strain is lower by about 5%. The corresponding change with temperature of critical strain $\varepsilon_{11}^{cr}$ of (24, 6) SWCNT as well as (14, 14) and (25, 0) SWCNTs are illustrated in Figure 8b. From the results, the effect of variation of temperature from 4.3 K to 500 K on critical strain is very limited, in other words, IPRs seem to be insensitive to temperature changes.

## 4. Conclusions

In this work, we discussed in detail the effect of chirality on the EPR, rigidity and IPR of SWCNT under different loading and revealed the auxetic mechanism of SWCNT above critical strain based on an MD simulation. First of all, we obtained the minimum EPR $\upsilon_{12}^{T}$ of 0.0385 and $\upsilon_{21}^{T}$ of 0.0284 for (4, 1) SWCNTs in theory. Then, we observed that the distribution and trend of EPRs $\upsilon_{12}^{T}$ and $\upsilon_{21}^{T}$ are sensitive to chiral index *m* when the diameter is small. However, this sensibility would be reduced with the increase of diameter. The insight of the regularity of EPR on *m* was partly revealed by IPE representing resistance to deformation. EPR is mainly affected by chiral angle instead when the diameter is greater than 20 Å. Secondly, we noticed that a SWCNT under different load mode could have an exactly opposite EPR trend, so that armchair SWCNTs under tension own the greatest $\upsilon_{12}^{T}$ but smallest $\upsilon_{21}^{T}$, while under compression, zigzag structure has the greatest $\upsilon_{12}^{C}$ but smallest $\upsilon_{21}^{C}$. Importantly, after obtaining the ratios of Young’s moduli and EPR for various chirality, we verified Maxwell’s theorem (*E*_11_ *υ*_21_= *E*_22_
*υ*_12_) and found that most SWCNTs appear anisotropic characteristics in mechanics. Finally, IPRs for both $\upsilon_{12}^{in}$ and $\upsilon_{21}^{in}$ of SWCNT would be negative once tensile strain exceeds critical strain. The minimum of critical strain $\varepsilon_{11}^{cr}$ and $\varepsilon_{22}^{cr}$ of zigzag or armchair SWCNT are both 6%, close to those of zigzag or armchair graphene. The $\varepsilon_{11}^{cr}$ and $\varepsilon_{22}^{cr}$ are also influenced by chiral angle and the greatest value can approach 0.18. The comparison of the contributions to IPR from two different deformation modes, bond stretch and angle variation, can give a good explanation of different critical strains for SWCNTs.

## Figures and Tables

**Figure 1 materials-15-08720-f001:**
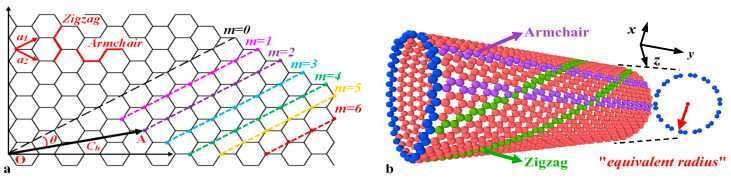
Schematic of a chiral SWCNT. (**a**) Schematic of the chiral parameters in graphene. Chiral vector ***C*_h_** for different chiral indices *m* is shown. (**b**) Schematic of equivalent radius in the *x*-*z* plane for a chiral SWCNT.

**Figure 2 materials-15-08720-f002:**
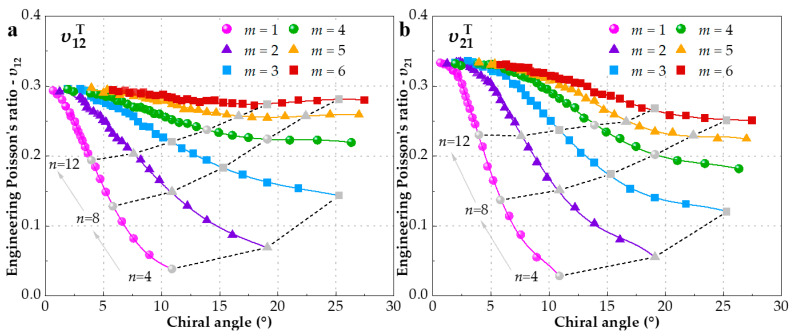
EPRs variation along with chiral angle. SWCNTs are grouped by chiral index *m* and their diameters are not restricted. (**a**) $\upsilon_{12}^{T}$ obtained from tension in longitudinal direction (**b**) $\upsilon_{21}^{T}$ obtained from tension in radial direction.

**Figure 3 materials-15-08720-f003:**
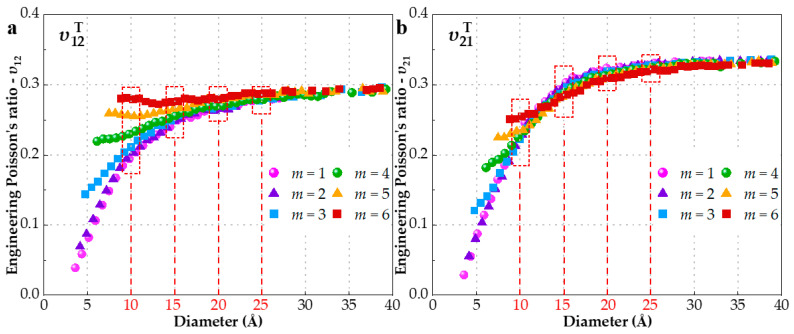
EPRs variation with respect to the diameter. SWCNTs are grouped by chiral index *m* and SWCNTs with the diameter of 10 Å, 15 Å, 20 Å, 25 Å are labeled in red. (**a**) $\upsilon_{12}^{T}$ (**b**) $\upsilon_{21}^{T}$.

**Figure 4 materials-15-08720-f004:**
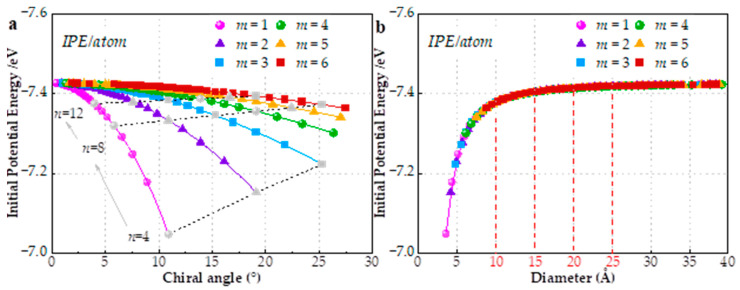
The initial potential energy of SWCNTs corresponding to Figure 2 (after relaxation and before stretch) (**a**) IPE differs according to chiral angle (**b**) IPEs differs according to diameter.

**Figure 5 materials-15-08720-f005:**
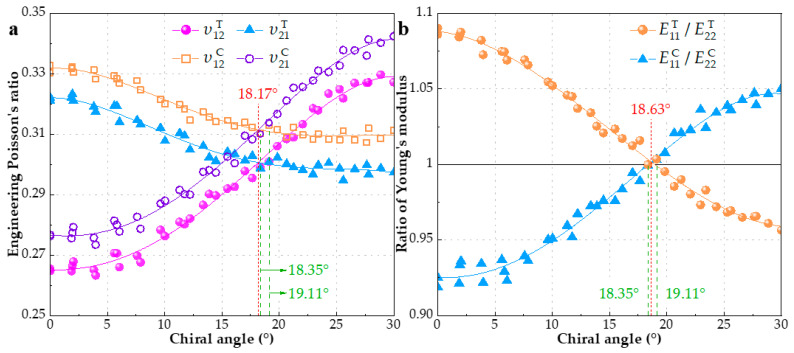
EPRs and ratio of Young’s moduli of 40 SWCNTs with diameter of 20 ± 1 Å are compared according to chiral angle (**a**) *υ*_12_ vs. *υ*_21_ under tension and compression (**b**) ratio of Young’s moduli under tension and compression.

**Figure 6 materials-15-08720-f006:**
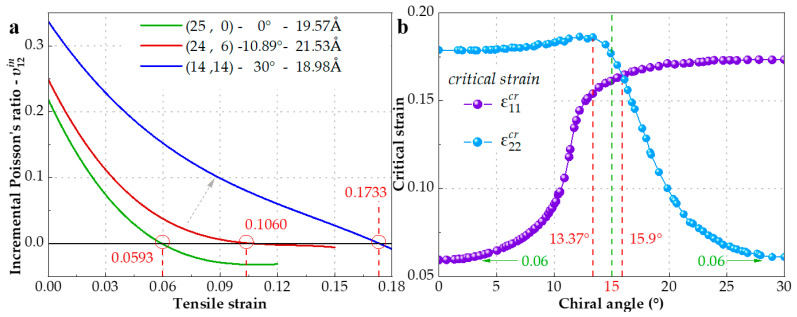
(**a**) The IPRs $\upsilon_{12}^{in}$ of three SWCNTs with 0° (zigzag), 10.89°, and 30° (armchair) chiral angle. The values of their critical strains are given and highlighted in red. (**b**) The critical strain of the SWCNTs with various chiral angle between 0° and 30°.

**Figure 7 materials-15-08720-f007:**
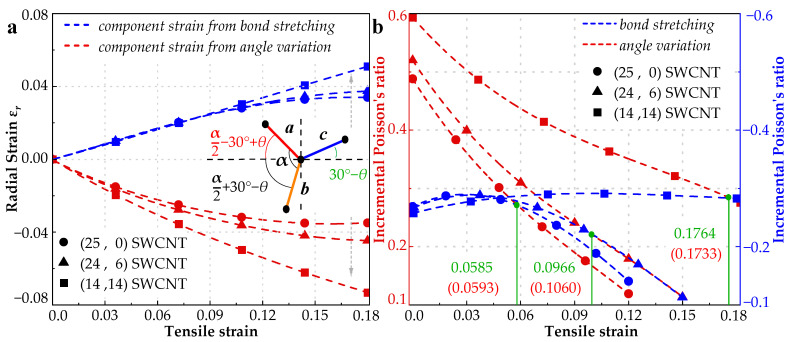
(**a**) Radial strain *ε_r_* owing to bond stretching and angle variation corresponding to chiral 0°(zigzag), 10.89°, and 30°(armchair). (**b**) IPRs owing to bond stretching and angle variation and the tensile strain of the intersection points (green) are compared with the critical strains (red) shown in Figure 6a.

**Figure 8 materials-15-08720-f008:**
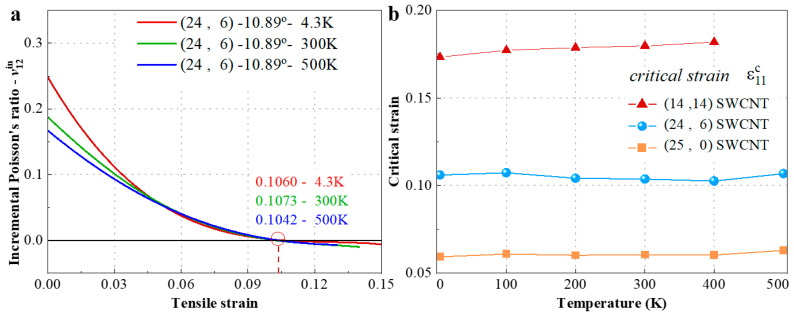
(**a**) The IPRs $\upsilon_{12}^{in}$ of (24, 6) SWCNT under the temperatures of 4.3 K, 300 K, and 500 K. (**b**) Variation of the critical strain with temperature for three SWCNTs (0°, 10.89°, and 30°).

**Table 1 materials-15-08720-t001:** EPRs and IPEs of SWCNTs with minimum diameter for a certain *m* (*m* = 1 to 6).

(*n*, *m*)	(4, 1)	(4, 2)	(4, 3)	(5, 4)	(6, 5)	(7, 6)
Chiral angle	10.89°	19.11°	25.28°	26.33°	27.00°	27.46°
Diameter (Å)	3.59	4.14	4.76	6.11	7.47	8.82
EPR $\upsilon_{12}^{T}$	0.0385	0.0694	0.1436	0.2194	0.2591	0.2802
EPR $\upsilon_{21}^{T}$	0.0284	0.0556	0.1206	0.1817	0.2247	0.2508
IPE (eV)	−7.048	−7.152	−7.223	−7.301	−7.341	−7.364

**Table 2 materials-15-08720-t002:** In 10.89° chiral angle family, $\upsilon_{12}^{T}$ and IPEs acquired with AIREBO and REBO potentials.

(*n*, *m*)	(4, 1)	(8, 2)	(12, 3)	(16, 4)	(20, 5)	(24, 6)
Diameter (Å)	3.59°	7.18°	10.76°	14.35°	17.94°	21.53°
EPR(AIREBO)	0.0385	0.1483	0.2065	0.2332	0.2491	0.2588
IPE (AIREBO, eV)	−7.048	−7.333	−7.384	−7.402	−7.410	−7.415
EPR (REBO)	0.0518	0.1678	0.2211	0.2511	0.2693	0.2806
IPE (REBO, eV)	−6.899	−7.260	−7.333	−7.359	−7.372	−7.379

**Table 3 materials-15-08720-t003:** EPRs and rigidities of distinctive SWCNTs satisfying $\upsilon_{12}^{T}$ = $\upsilon_{21}^{T}$ and $E_{11}^{T}$ = $E_{22}^{T}$ under tension.

Group	(*n, m*)	*θ* (°)	D (Å)	$\upsilon_{12}^{T}$	$\upsilon_{21}^{T}$	△*υ* *	$C_{11}^{T}$ (TPa·nm)	$C_{22}^{T}$ (TPa·nm)	△*C* **
G25	(26, 11)	16.83°	25.76	0.3070	0.3075	−0.16%	0.2961	0.2954	0.24%
	(24, 11)	17.90°	24.27	0.3072	0.3061	0.36%	0.2957	0.2966	−0.30%
G20	(19, 9)	18.35°	19.38	0.2991	0.2987	0.13%	0.2983	0.2984	−0.03%
	(20, 10)	19.11°	20.71	0.3009	0.3007	0.07%	0.2965	0.2955	0.34%
G15	(14, 7)	19.11°	14.50	0.2877	0.2873	0.14%	0.3040	0.3049	−0.30%
G10	(10, 3)	12.73	9.23	0.2043	0.2046	−0.15%	0.3338	0.3343	−0.15%
	(11, 4)	14.92	10.53	0.2341	0.2343	−0.09%	0.3231	0.3242	−0.34%

* △*υ* = [( $\upsilon_{12}^{T}$ − $\upsilon_{21}^{T}$)/$\upsilon_{12}^{T}$] × 100%. ** △*C* = [($C_{11}^{T}$ − $C_{22}^{T}$)/$C_{11}^{T}$] × 100%.

**Table 4 materials-15-08720-t004:** EPRs and rigidity of SWCNTs satisfying $\upsilon_{12}^{C}$ = $\upsilon_{21}^{C}$ and $E_{11}^{C}$ = $E_{22}^{C}$ under compression.

Group	(*n, m*)	*θ* (°)	D (Å)	$\upsilon_{12}^{C}$	$\upsilon_{21}^{C}$	△*υ* *	$C_{11}^{C}$ (TPa·nm)	$C_{22}^{C}$ (TPa·nm)	△*C* **
G25	(26, 11)	16.83°	25.76	0.3174	0.3162	0.38%	0.3017	0.3026	−0.30%
	(24, 11)	17.90°	24.27	0.3168	0.3177	−0.28%	0.3029	0.3019	0.33%
G20	(19, 9)	18.35°	19.38	0.3103	0.3100	0.10%	0.3047	0.3049	−0.07%
	(20, 10)	19.11°	20.71	0.3130	0.3138	−0.26%	0.3042	0.3033	0.30%
G15	(14, 7)	19.11°	14.50	0.2943	0.2945	−0.07%	0.3119	0.3122	−0.10%
G10	(10, 3)	12.73	9.23	0.2134	0.2141	−0.33%	0.3418	0.3429	−0.32%
	(11, 4)	14.92	10.53	0.2404	0.2408	−0.17%	0.3340	0.3336	0.12%

* △*υ* = [($\upsilon_{12}^{C}$ − $\upsilon_{21}^{C}$)/$\upsilon_{12}^{C}$] × 100%. ** △*C* = [($C_{11}^{C}$ − $C_{22}^{C}$)/$C_{11}^{C}$] × 100%.

## Data Availability

Not applicable.

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
