# Peer review of "Chirality-Dependent and Intrinsic Auxeticity for Single-Walled Carbon Nanotubes"

_materials, 2022, doi:10.3390/ma15248720_

Round 1
Reviewer 1 Report
The authors present a thorough analysis of the effect of chirality and diameter on Poisson's ratio, incrimental Poisson's ratio, and stiffness of single-walled carbon nanotubes (SWCNTs) subjected to axial and radial tension/compression.
Molecular dynamics based on the AIREBO potential is the main research tool in this work.
The degree of novelty of this work is not high, since similar problems were solved using different approaches and the main trends noted in this work have already been described. Nevertheless, I find this systematic work useful because the results are well illustrated and explained. The authors report the effect of chirality and strain on the negative incrimental Poisson's ratio.
Therefore, I recommend this work for publication in Materials after the authors make minor edits.
Minor issues:
It would be great if the authors consulted with native English speakers and improved the quality of the English throughout the manuscript.
I can't accept the first sentence of the abstract. It is generally believed that the excellent mechanical properties of CNTs are due to the particularly strong C-C covalent bond, with chirality being a minor factor.
line 11: single-walled carbon nanotubes -> SWCNTs
line 60 - thoroughly
line 77: add "was" in front of "also reported by Ref. [13, 24]."
line 132: index -> indices
line 213: chiral -> chirality or chiral angle
line 217: After 20 Å, -> For diameters above 20 Å,
line 240: achieves of -> reaches the value of
line 255: form -> from
line 256: the question mark is missing?
lines 255,256: Since increasing m or n leads to the same result of larger diameter, what happens if the diameter of SWCNT is enough big.
Since increasing m or n has the same result of increasing the diameter, what happens if the diameter of the SWCNT is large enough?
Equation (12): make arcsine not italic
line 361: at critical strain -> above critical strain
Please note that plural of Young's modulus is Young's moduli
ratios of Young's modulus -> ratios of Young's moduli
line 371: sentence has no grammatical sense and its meaning is unclear
The following two works can be included in the review of the existing literature on the topic studied in this manuscript:
E.A. Korznikova, et al., Partial auxeticity of laterally compressed carbon nanotube bundles. Physica Status Solidi - Rapid Research Letters, 2022, 16(1), 2100189.
L.K. Galiakhmetova, et al., Negative thermal expansion of cCarbon nanotube Bbundles. Physica Status Solidi - Rapid Research Lettersthis link is disabled, 2022, 16(3), 2100415
Reviewer 2 Report
In this manuscript, the authors reported molecular dynamics (MD) simulation to analyze the engineering Poisson’s ratio (EPR) and incremental Poisson’s ratio (IPR) of single-walled carbon nanotubes with different chiral angles respectively. It was found that the new observation showed two exactly opposite trends of EPR existing not only in tension and compression but also in a longitudinal and radial direction. Moreover, they show the critical strain, over which SWCNT would be auxetic, was also chirality-dependent. It is a well-written paper in general, and the characterization is in-depth implemented. Therefore, I recommend a minor revision of the current manuscript for publication in the journal of Materials. The following points should be addressed:
1. The graphical abstract should be presented by the scheme. It will improve the quality of the manuscript.
2. In the abstract, please provide more quantitative information
3. What are the innovation and prospects for this work? Please, clearly indicate in the introduction part.
4. Based on Figure 7, the authors stated “bond stretching leads to positive radial strain resulting in negative Poisson’s ratio, while angle variation contributes negative radial strain resulting in positive Poisson’s ratio”. Please extend discussion on this claim and further support from the literature
5. The advantages of engineering Poisson’s ratio (EPR) processed materials over other methods should be provided.
6. The writing style, grammar and language usage should be checked by a native speaker.
Reviewer 3 Report
The authors study auxeticity of SWCNTs using MD simulation. in my opinion the topic and related results are interesting and worthy of publication. However, some points remain unclear and should be addressed before recommending for publication.
1. The main concern of mine is the effect of temperature on the PR which is undeniable and important. There is no discussion can be seen on this. However, presence of vacancies can be effective and should be analyzed to reach general conclusion. however, temperature is more important and depending of the authors’ sources some sections better to be added.
2. It’s better to manage the reference list and add some references on auxetic nanostructures and various studies on CNTs to enrich the introduction and cover wide range of works on CNTs:
For CNTS: “ Applied Physics A 121.1 (2015): 223-232.”
For Auxeticity: “Diamond and Related Materials 124 (2022): 108956.” And etc.
Reviewer 4 Report
In the present manuscript, the authors study the effect of the chirality of the Single-walled carbon nanotubes (SWCNTs) on their mechanical properties such as engineering Poisson’s ratio (EPR) and incremental Poisson’s ratio (IPR). They have employed molecular dynamics simulation techniques in their study. In this study, they investigated the effect of chirality on the mechanical properties of SWCNT and show that EPR is dependent on the chiral index, m. In addition, they provide some evidence of the negative IPR value for SWCNT and their auxetic behavior.
The paper is well written, the model is clearly explained, and the numerical analysis appears to be technically sound and carefully performed.
I have no significant criticism and believe the manuscript could be published in Materials in its present form.
Author Response
Thank you very much for your positive comments and recognition of our work.
Round 2
Reviewer 3 Report
The Authors addressed my concerns properly.